# Characterization of Primary Mucinous Ovarian Cancer by Diffusion-Weighted and Dynamic Contrast Enhancement MRI in Comparison with Serous Ovarian Cancer

**DOI:** 10.3390/cancers15051453

**Published:** 2023-02-24

**Authors:** Laretta Grabowska-Derlatka, Pawel Derlatka, Marta Hałaburda-Rola

**Affiliations:** 1Second Department of Clinical Radiology, Medical University of Warsaw, Banacha 1a St., 02-097 Warsaw, Poland; 2Second Department Obstetrics and Gynecology, Medical University of Warsaw, Karowa 2 St., 00-315 Warsaw, Poland

**Keywords:** magnetic resonance diffusion-weighted imaging (DWI), magnetic resonance dynamic contrast enhancement (DCE), mucinous ovarian cancer (MOC), low-grade serous ovarian cancer (LGSC), high-grade serous ovarian cancer (HGSC)

## Abstract

**Simple Summary:**

Epithelial ovarian cancer (EOC) has always been considered one of the most challenging problems for gynecologic oncologists. This is not only due to difficulties in treatment, but also in the diagnosis and differentiation of this malignant tumor. MR imaging techniques, combined with diffusion-weighted imaging (DWI) or dynamic contrast enhancement (DCE), make it possible not only to identify the exact location of lesions, but also to identify the types of EOC. In our study, we described the characteristics of a rare ovarian malignancy, mucinous ovarian cancer (MOC), on MRI. We compared the DWI and DCE values for MOC and more or less aggressive histological types of EOC, such as high-grade serous carcinoma (HGSC) (type I and type II) as well as the low-grade serous carcinoma (LGSC) (both type I).

**Abstract:**

(1) Background. The purpose of this study is to evaluate the diagnostic accuracy of a quantitative analysis of diffusion-weighted imaging (DWI) and dynamic contrast enhanced (DCE) MRI of mucinous ovarian cancer (MOC). It also aims to differentiate between low grade serous carcinoma (LGSC), high-grade serous carcinoma (HGSC) and MOC in primary tumors. (2) Materials and Methods. Sixty-six patients with histologically confirmed primary epithelial ovarian cancer (EOC) were included in the study. Patients were divided into three groups: MOC, LGSC and HGSC. In the preoperative DWI and DCE MRI, selected parameters were measured: apparent diffusion coefficients (ADC), time to peak (TTP), and perfusion maximum enhancement (Perf. Max. En.). ROI comprised a small circle placed in the solid part of the primary tumor. The Shapiro–Wilk test was used to test whether the variable had a normal distribution. The Kruskal–Wallis ANOVA test was used to determine the *p*-value needed to compare the median values of interval variables. (3) Results. The highest median ADC values were found in MOC, followed by LGSC, and the lowest in HGSC. All differences were statistically significant (*p* < 0.000001). This was also confirmed by the ROC curve analysis for MOC and HGSC, showing that ADC had excellent diagnostic accuracy in differentiating between MOC and HGSC (*p* < 0.001). In the type I EOCs, i.e., MOC and LGSC, ADC has less differential value (*p* = 0.032), and TTP can be considered the most valuable parameter for diagnostic accuracy (*p* < 0.001). (4) Conclusions. DWI and DCE appear to be very good diagnostic tools in differentiating between serous carcinomas (LGSC, HGSC) and MOC. Significant differences in median ADC values between MOC and LGSC compared with those between MOC and HGSC indicate the usefulness of DWI in differentiating between less and more aggressive types of EOC, not only among the most common serous carcinomas. ROC curve analysis showed that ADC had excellent diagnostic accuracy in differentiating between MOC and HGSC. In contrast, TTP showed the greatest value for differentiating between LGSC and MOC.

## 1. Introduction

Epithelial ovarian cancer (EOC) is one of the most common malignancies in women worldwide. Due to the lack of early clinical symptoms and screening options, it ranks fourth among causes of cancer deaths worldwide [1,2]. EOC has been divided into two types depending on molecular and clinicopathological factors. Type I is less common and includes subtypes such as low-grade serous carcinoma (LGSC), mucinous ovarian carcinoma (MOC), endometrioid carcinoma and clear cell carcinoma. Type II is high-grade serous carcinoma (HGSC) and is the leading malignant tumor of the ovary [3,4]. The histologic type of EOC, optimal cytoreductive surgery and platinum-based, adjuvant chemotherapy are considered prognostic factors [5,6]. MOC is a rare type of ovarian cancer, with an estimated incidence of 3–12% of all EOCs. Typically, MOC grows slowly and is confined to the ovary at the time of diagnosis [7,8,9,10,11]. It is the most common type of EOC in women under 40 years of age [12]. According to the National Cancer Institute, 51% of HGSC cases are diagnosed in stage III and in 29% in stage IV, according to FIGO [13]. In MOC, 80% of cases are recognized in stage I, according to FIGO [14]. In the early clinical stages, MOC has a better prognosis than HGSC, while in advanced cases, the prognosis is worse, due to poor response to platinum-based chemotherapy [15].

According to the European Society of Uro-Genital Radiology (ESUR) guidelines, preoperative staging of EOC includes contrast enhanced computed tomography (CT) of the chest, abdomen and pelvis [16]. Magnetic resonance imaging (MRI) can provide more comprehensive information than CT on diagnosis of EOC, especially about the local extent of the disease. Compared with CT, MR diffusion-weighted imaging (DWI) combined with apparent diffusion coefficients (ADC) has shown promise not only in tumor staging, but also in assessing tumor type and predicting the clinical course of the disease [17,18,19]. Dynamic contrast enhanced (DCE) MRI improves the diagnostic accuracy of conventional MRI. In addition, there are studies that DCE can be useful in differentiating between highly and low aggressive EOC [20]. Furthermore, some studies have proposed the use of perfusion MRI as a prognostic tool in EOC [20]. However, these studies concerned serous carcinomas and the differentiation of its types, LGSC and HGSC. To date, there are few papers describing the morphological features of MOC in MRI, with little attention paid to DCE and DWI parameters [21].

Better insight into tumor morphology prior to surgical treatment enables easier and more precise planning of the scope of the operation. In tumors considered chemoresistant or poorly responsive to chemotherapy, such as MOC and low-grade serous ovarian cancer, primary optimal cytoreduction, precisely R-0 cytoreduction, is essential for prognosis. In these cases, suboptimal cytoreductive surgery or the use of neoadjuvant chemotherapy yields significantly worse results. The information obtained in the preoperative MRI that we are dealing with EOC types resistant to systemic treatment can be beneficial for the patient.

The purpose of our study is to evaluate the diagnostic accuracy of DWI and DCE MRI quantitative analysis of the primary tumor in MOC patients and in differentiating between serous EOC (LGSC and HGSC) and MOC. The analysis of the primary tumor is dictated by the fact that MOC is diagnosed in early stages more often than other types of EOC, especially HGSC. In addition, we analyze selected DWI and DCE parameters for differentiating between serous EOC (LGSC and HGSC) and MOC.

## 2. Material and Methods

### 2.1. Study Protocol

This single-center prospective study was conducted in the Second Department of Obstetrics and Gynecology and in the Second Department of Clinical Radiology at the Medical University of Warsaw. Patients with clinical suspicion of ovarian cancer based on CT or transvaginal ultrasound were included in the study.

The exclusion criteria were current treatment of coexisting cancers and contraindications to MRI with gadolinium contrast administration. Histological grade and type of cancer were assessed according to the 2014 WHO criteria. The following types of EOCs were analyzed in the study: LGSC, HGSC and MOC [18]. The profile of immunohistochemical (IHC) staining was as follows: CK7, CK20, PAX8, CDX2 WT1, MUC2, ER, and PR. The FIGO criteria (International Federation of Obstetricians and Gynecologists) were applied to disease staging. The patient characteristics, and clinical and histopathological data are collected in Table 1.

### 2.2. MRI Protocol

MRI data were acquired with a 1.5T MR scanner (MAGNETOM Avanto, Siemens AG, Erlangen, Germany).

The MRI protocol applied in the study included sequences as follows: turbo spin-echo (TSE), T2 weighted images (T2WI), fat suppressed T2-weighted images (fsT2WI), turbo inversion recovery magnitude (TIRM), diffusion-weighted echoplanar imaging (DW-EPI), and pre- and postcontrast dynamic T1-weighted gradient echo sequences (3D T1-GRE). Details of the applied parameters of MR imaging are presented in Table 2.

The axial DW images were acquired using unified multi-slice EPI sequence 30 mm × 6 mm slices (pelvic part); 360 mm × 360 mm FoV; TR = 4250 ms; TE = 73 ms; and diffusion weights of 0, 50, 500, 1000, 1500 mm^2^/s. The parameters are collected in Table 2. Motion correction was completed automatically.

The MRI were analyzed by two board-certified radiologists experienced in pelvis MRI (one with more than 15 years of experience in oncologic imaging and a board-certified radiologist with a European Diploma in Radiology certificate).

The regions of interest (ROI) were drawn on the apparent diffusion coefficient (ADC) maps and all b values DWI outlined in the Multimodality Workplace Station (GE AW Server 3.2 ext. 4.0, Volume Viewer 16.0 Ext. 2 Ready View).

Each radiologist examined each patient twice. To compare the ADC values recorded by the two radiologists and for the purpose of analyzing these values, they are referred to as follows. ADC1 and ADC2 correspond to the examination performed by the first radiologist, and ADC3 and ADC4 correspond to the examination performed by the second radiologist.

On all the DWI (with b values of 0, 50, 500, 1000, 1500 mm^2^/s), the ROI contained a small circle with a diameter of 5mm that was placed in the solid part of the primary tumor, avoiding partial volume effect, areas of necrosis and artifacts. The ROIs were duplicated from the DWI to the corresponding ADC maps and the measurement on the ADC was recorded. The T1WI (non-contrast and post-contrast) and dynamic contrast enhancement (DCE) sequence parameters for dynamic analysis are collected in Table 2.

The ROIs were placed on the post contrast DCE images and replicated to DCE parametric maps. During DCE image acquisition, non-contrast images were acquired first, followed by contrast agent administration and continued acquisition. On the DCE images, parameters such as perfusion maximum enhancement (Perf. Max En.) and time to peak (TTP) were measured. The DCE parametric maps were generated automatically using Workplace Station. In all patients, gadobutrol (Gadovist, Bayer AG, Leverkusen, Germany) was administered as a bolus dose of 0.1 mmol/kg, followed by a bolus dose of 20 mL of physiological saline (NaCl 0.9%).

### 2.3. Statistical Analysis

IBM SPSS Statistics (version: 28.0.1.0(142)) was used to analyze the distribution of variables, perform statistical tests, as well as calculate statistics. PQStat (version: 1.8.4.) was used for the same purpose to analyze the data and to prepare all the visualizations seen in this article. The Shapiro–Wilk test was used to examine if a variable is normally distributed. During the analysis, it was decided to divide patients into three groups by EOC diagnosis (LGSC, HGSC and MOC patients) in order to compare the parameters for these groups when examined by a radiologist. To determine the *p*-value needed to compare median values of interval variables, the Kruskal–Wallis ANOVA test was used. The test is used specifically for independent variables that do not meet the condition of normal distribution. The main aim was to define if the *p*-value is less than 0.05, which indicates a statistically significant difference. Based on the post hoc (Conover–Iman) analysis, the two homogeneous groups were created based on similar median values. For the interval variables, to determine the inter-observer agreement between the two reviewers, the interclass correlation coefficient (ICC) was used. The determination of the ROC curve and the calculation of the area under this curve (AUC), as well as the calculation of sensitivity and specificity were used to compare which of the three quantitative parameters could give the best result in assigning patients to the appropriate groups. Due to the fact that there were three groups defined in the study while two groups are needed to plot the ROC curve, it was decided to assign patients to two groups for this analysis: LGSC with MOC and HGSC with MOC.

## 3. Results

The study included 66 women aged 33–78 years (median 57.5 (48.5–64), interquartile range (IQR) = Q3 − Q1 = 15.5), who were diagnosed with EOC on the final histopathological examination.

### 3.1. Primary Tumor DWI

Patients were examined using ADC, where two radiologists examined each patient twice. ADC1 and ADC2 correspond to the examination performed by the first radiologist, and ADC3 and ADC4 correspond to the examination performed by the second radiologist.

The Shapiro–Wilk test showed that the variables ADC1, ADC2, ADC3 and ADC4 were not normally distributed (*p* value < 0.05).

The calculated median values for ADC1, ADC2, ADC3 and ADC4 for the three defined groups are presented in Figure 1.

Due to the small differences in ADC values, it was decided to verify the inter-observer agreement using the ICC.

### 3.2. Inter-Observer Agreement

The ICC was used to examine the inter-observer agreement for the quantitative parameter, ADC. For the ADC parameter, 132 (2 radiologists × 66 patients) observations were included. Based on the ICC values, there was an excellent, statistically significant inter-observer agreement between the two observers in assessing quantitative ovarian involvement by ADC. The inter-observer concordance oscillated at the level of excellent concordance, with ICC > 0.9, 95% CI: 0.899–0.961; *p* < 0.000001 (Figure 1).

Figure 2 and the box plot in Figure 2 show the results for ADC, treating the ADC parameter as one study, without dividing it into ADC 1–4.

### 3.3. Primary Tumor DCE

Similar to the statistical analysis of ADC 1–4 values, the normal distribution of the parameters TTP and Perf. Max. En. was tested with the Shapiro–Wilk test and a *p*-value < 0.05 was obtained. The test showed that there was no normal distribution, so the median was used in Figure 3 instead of the arithmetic mean. The statistical analysis was also performed across the three groups: LGSC, HGSC and MOC.

The *p*-value < 0.05 by the Kruskal–Wallis ANOVA test indicates that the differences in the TTP and Perf. Max. En. parameters between the groups of patients according to EOC diagnosis were statistically significant. Thanks to the post hoc test, it was noted that the LGSC and HGSC patients had similar values for the parameters TTP and Perf. Max. En., which were statistically different from those for patients with MOC. For the TTP parameter, the LGSC and HGSC patients had lower median values (292 and 330, respectively) than the MOC patients (410), *p*-value = 0.009. However, for the Perf. Max. En., the median value for the patients in the LGSC = 260 and HGSC = 230 groups is higher than that of the patients with MOC = 141, *p*-value = 0.004.

The median values obtained for the TTP and Perf. Max. Enh. parameters are shown as box plots (Figure 3a,b).

Figure 4, Figure 5, Figure 6, Figure 7, Figure 8, Figure 9 and Figure 10 show the MOC morphology in T1-weighted and T2-weighted presentations and differences between MOC, LGSC and HGSC in DWI and DCE.

### 3.4. ROC Curve for LGSC vs. MOC

Figure 4 shows all the values important for the comparison and analysis regarding the ROC curve for the three quantitative parameters, i.e., range of values, sensitivity, specificity, cut-off value, AUC value and *p*-value.

Using the ROC curves (Figure 11, Figure 12 and Figure 13), the optimal cut-off value was selected that distinguishes LGSC patients from MOC patients. Comparing the results obtained for the ROC curves, it can be seen that the cut-off value for ADC is ≥1382, that for TTP is ≥354, and for maximum perfusion enhancement it is ≤142. Based on these results, it can be concluded that for ADC and TTP, patients who achieved scores of 1382 and 354 or higher, respectively, should be assigned to the MOC group, whereas for Perf. Max. En., patients with values 142 or less are assigned to the MOC group.

The three ROC curves showed good, excellent and very good diagnostic validity for the three measured parameters. The AUC values were good for ADC (0.744), excellent for TTP (0.9), and very good for Perf. Max. En. (0.814). The sensitivity oscillated at the levels of 66.7%, 100%, and 66.7%, respectively, and the specificity was at the level of 100%, 80% and 100%, respectively. Based on the obtained AUC values, we can conclude that LGSC is best differentiated from MOC by TTP, followed by Perf. Max. En. and ADC.

### 3.5. ROC Curve for HGSC vs. MOC

By comparison with the previously defined ROC curves, it was determined which parameter (ADC, TTP or Perf. Max. En.) can better assign patients to the HGSC or MOC groups. Thus, the two groups were determined (HGSC and MOC).

Figure 5 shows all the values important for the comparison and analysis regarding the ROC curve for the three quantitative parameters, i.e., range of values, sensitivity, specificity, cut-off value, AUC value and *p*-value.

By using ROC curves (Figure 14, Figure 15 and Figure 16), the optimal cut-off value was determined, that best divides the study population into two groups: the patients with HGSC and the patients with MOC. Comparing the results obtained for the ROC curves, it can be seen that the cut-off value for the ADC parameter was ≥1028.15, and for the TTP it was ≥354 (the same value as when the patients were divided into the LGSC and MOC groups), and for the Perf. Max. En. it was ≤142 (also the same value as in the analysis of patients by LGSC and MOC groups). Based on these results, it can be concluded that for the ADC and TTP parameters, the patients who score 1028.15 and 354 or higher, respectively, should be assigned to the MOC group, whereas for Perf. Max. En., patients with scores 142 or less are assigned to the MOC group.

The three ROC curves showed excellent, good and very good diagnostic validity for the three measured parameters. The AUC values were as follows: ADC, excellent (0.932); TTP, good (0.722); and Perf. Max. En., very good (0.795). The sensitivity was 83.3%, 100% and 53.8%, respectively, and the specificity was 94.9%, 53.8% and 100%, respectively.

Based on the obtained AUC values, we can conclude that HGSC is best differentiated from MOC by ADC, followed by Perf. Max. En. and TTP.

## 4. Discussion

The results presented enabled us to determine the characteristics of MOC in DWI and DCE MRI. Our study showed that the use of parameters such as ADC, TTP and Perf. Max. En. make it possible to differentiate MOC and serous EOC and to divide the latter into HGSCs and LGSCs.

The postoperative diagnosis of MOC is based on microscopic examination and immunohistochemical staining (IHC staining). MOC shares common features with other mucinous tumors, including gastrointestinal metastases, such as positive CK20, CEA, Ca19-9 and CDX2 staining. However, the primary differentiating factor is CK7 positivity. The standard IHC profile for MOC is CK7+, CK20+/−, CDX2+/−, PAX8-, WT1-, ER-, PR- and SATB2 [22,23]. In our study, the IHC profile was similar and was as follows: CK7, CK20, CDX2, PAX8, WT1, ER, PR, and MUC-2.

MRI is the imaging modality with the highest tissue resolution. It is the method of choice in the differentiation of primary ovarian tumors as well as in the diagnosis of metastatic lesions [24,25,26,27].

In our study, we placed special focus on evaluating quantitative parameters of DWI and DCE. Nevertheless, qualitative analysis of the tumor was also included in our study as the two methods seem to be complementary to each other.

The morphological elements of the primary tumor are taken into consideration in the differential diagnosis. Epithelial ovarian tumors contain solid components such as thick septa and internal solid components, while the content of solid elements in mucinous tumors is low. When determining the degree of malignancy in mucinous tumors, the size of the tumor is taken into account, especially above 10 cm, as well as the number of solid elements, mainly those with the “honeycomb” appearance [21,27,28].

Some morphologic features of fluid content, such as a high signal on T1WI with fat suppression or an intermediate signal on T2WI, suggesting the presence of mucus, may raise the suspicion of a mucinous tumor. However, these features are exclusive to mucinous tumors. In serous carcinomas, mucous contents may also be present among fluid components [21,27]. Large size of a tumor, unilateral mass, lack of spread to the peritoneum and infiltration of adjacent tissues may suggest mucinous carcinoma but are still not sufficient to make a diagnosis [21,27].

Morphological features are not sufficient to determine the type and the grade of malignancy of an ovarian tumor. Additional information may be provided by the DWI as well as the DCE curves in the dynamic post-contrast examination and, as emphasized by many authors, can be essential in differential diagnosis [19,20,29,30].

In our material, we focused on the comparative quantitative evaluation of DCE and DWI MRI in differentiating mucinous carcinomas from LGSC and HGSC. It has been demonstrated that differentiation of LGSC from HGSC is possible based on DWI and ADC [19,31,32]. Some authors claim that certain perfusion parameters, such as k-tTRANS and TTP, may be considered as differentiating and prognostic factors for serous carcinomas simultaneously [20,32,33].

Currently, there are no reports on the differentiation of MOC and other types of EOC based on diffusion and perfusion values on MRI. Single studies have described differences between MOC and mucinous borderline ovarian tumors (MBOT). The authors report that within the solid parts of the tumor, ADC values in MOC are lower than in MBOT [34,35]. These findings are similar to studies on the serous EOCs, i.e., LGSC and HGSC. HGSCs, being more aggressive and less differentiated, show greater diffusion restriction and lower ADC values compared with well-differentiated LGSCs. This is also confirmed by the negative correlation with IHC markers of proliferation, such as Ki67 [19,32,33]. Our study suggests that MOC is associated with LGSC in terms of DWI and DCE values. This is consistent with the division of EOC into two molecular types. Type I includes LGSC, MOC, endometrioid and clear cell carcinoma, while HGSC is classified as the more aggressive type II. LGSC and MOC are associated with BRAF and KRAS mutations. Mutation of p53 is characteristic of HGSCs [36]. The results shown in Figure 3 indicate statistically significant differences in median ADC values between MOC, LGSC and HGSC. However, the median values for type I (MOC and LGSC) are more similar than for HGSC (type II). This is also confirmed by analysis of the ROC curves for MOC and HGSC, showing that ADC has excellent diagnostic accuracy in differentiating between MOC and HGSC. In type I EOCs, i.e., MOC and LGSC, ADC has less differential value while TTP can be considered the most valuable parameter due to its excellent diagnostic accuracy. In this study, TTP was found to be the most valuable parameter.

Our study has some limitations. First, it is a single-center study. Secondly, MOC is a rare type of EOC (only 3–4%). Due to this fact, a small number of patients were enrolled in the study, but this problem concerns all authors describing this type of EOC.

## 5. Conclusions

DWI and DCE appear to be very good diagnostic tools in differentiating between serous carcinomas (LGSC, HGSC) and MOC. Significant differences in median ADC values between MOC and LGSC and between MOC and HGSC indicate the usefulness of DWI in differentiating between less and more aggressive types of EOC, not only in the group of most common serous carcinomas.

ROC curve analysis showed that ADC had excellent diagnostic accuracy in differentiating between MOC and HGSC. In contrast, TTP showed the greatest differentiating value when diagnosing between LGSC and MOC.

## Data Availability

The data presented in this study are available on request from the corresponding authors.

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
