# Peer review of "Characterization of Primary Mucinous Ovarian Cancer by Diffusion-Weighted and Dynamic Contrast Enhancement MRI in Comparison with Serous Ovarian Cancer"

_cancers, 2023, doi:10.3390/cancers15051453_

Round 1
Reviewer 1 Report
The authors report on the evaluation of the diagnostic accuracy of a quantitative
analysis of diffusion weighted imaging (DWI) and dynamic contrast enhanced (DCE) MRI of mucinous ovarian cancer (MOC) and to differentiate between low grade serous ovarian cancer
(LGSC), high-grade serous ovarian cancer (HGSC) and MOC in primary tumor.
I appreciate it is an interesting study, well conducted, with a rigorous investigation. Also, I consider the manuscript with a significant clinical impact. Good work!
Author Response
Dear Reviewer
Thank you for your review. We appreciate your positive opinion about our study very much. :)
Reviewer 2 Report
This manuscript, “Primary mucinous ovarian cancer. Characterization by diffusion-weighted and dynamic contrast enhancement MRI. Comparison with serous low grade and high grade ovarian malignancy, by Grabowska-Derlatka et al. is a well-controlled study to detect differences in pathology-confirmed mucinous vs serous (low–grade or high-grade) ovarian cancer. The study had two board-certified radiologists analyze each patient twice and the inter-observer agreement was assessed.
Major points:
1. This paper is arguing that MRI could be used to predict whether ovarian cancer tumors are mucinous, low-grade serous or high-grade serous. However, there is no discussion of how these predictions could improve patient care compared to the standard histopathological diagnosis. Would this earlier prediction alter choice of treatment options? What do the authors believe would be the benefit of MRI analysis? Please include this in the Introduction and/or Discussion. Also include in the Introduction the current methods for diagnosing MOC and how MRI would improve diagnosis.
2. ROC curves indicate the sensitivity and specificity of the prediction comparing MOC to LGSC and MOC to HGSC. Please place this in the context of other MRI tests that are used for this purpose. Is a very good diagnostic validity worthwhile pursuing or do you want to focus only on those that are excellent?
3. It would be ideal to have a second group of ovarian cancer patients that have not been subdivided by type where the MRI parameters are used to predict the ovarian cancer subtype to test this modality as a true diagnostic. If it is out of the scope of this paper, it should be mentioned in the Discussion as a future study to test the diagnostic potential to differentiate between MOC, LGSC and HGSC.
Minor Points:
1. The first sentence of the Results is unclear, “..who were diagnosed with primary EOC the final histopathological examination.” I believe something is missing here.
2. In the third paragraph of the Discussion, the first sentence is unclear, “Surgical treatment of MOC is likewise other types of EOC.” Please clarify this statement.
Author Response
Dear Reviewer
The authors would like to thank you for your critical insight into the manuscript and for the specific comments. The manuscript was edited to meet the points raised by the Reviewer:
Major point 1.
Undoubtedly, the diagnosis of ovarian cancer is made with the final histopathological examination. However, good preoperative characterization of the tumor leads to better preparation for the primary surgery. Optimal cytoreductive surgery, considered as R-0 surgery, is an important prognostic factor in chemoresistant or poorly responsive tumors such as MOC, low grade serous ovarian cancer. In such cases, suboptimal cytoreductive surgery or the use of neoadjuvant chemotherapy brings significantly worse results. The information obtained in MRI that we are dealing with EOC types that are resistant to systemic treatment is beneficial for the patient. The information was added to the introduction.
Major point 2.
Thank you for the comment. As we wrote in the introduction and discussion, nowadays the differential diagnosis of ovarian cancer in MRI is based on the qualitative assessment of the tumor. We believe that eachMRI protocol for ovarian tumor examination should include those parameters that will bring us possibly close to the initial diagnosis, therefore the parameters differentiating tumors in diffusion weighted imaging on ADC maps and perfusion parameters brings explicit added quantitative value.
Major point 3
This is how the study group was created in our work. The MRI examination performed before the surgery and comparison of the described diffusion and perfusion parameters with the postoperative histopathological examination were aimed at evaluating the DWI and DCE values as predictors of EOC type. In our study, we focused on the results of quantitative parameters. Of course, in the descriptions of preoperative MRI examinations, qualitative parameters (tumor morphology) are also taken into account, which also brings us closer to the diagnosis. We added such information in the discussion.
Minor point 1
The sentence was changed as follows: “The study included 66 women aged 33-78 years (median 57.5 [48.5; 64], interquartile range(IQR)=Q3-Q1=15.5), who were diagnosed with EOC on the final histopathological examination.”
Minor point 2.
The whole paragraph was removed as it brings no relevant information to the topic.

Reviewer 3 Report
Dear Authors,
Thank you very much for allowing me to express my opinions related to your work. As a researcher myself, I admire and respect the effort you put into constructing your study and building this manuscript.
Below, you can find my comments regarding certain issues. I hope these comments will help you improve both your current and future work.
Simple summary:
· the sentence “This makes it easier to develop a treatment strategy.” is too. The results of this study can, at best, increase confidence in the imaging diagnosis of MOC, but cannot decide the therapeutic course. Please remove this statement.
Abstract
· It should be mentioned in the abstract which area of the tumor was used to calculate the quantitative parameters. It may sound unnecessary, but some studies have measured these parameters both from the solid portion of the tumor and within the fluid portions. This should be clarified at the outset.
· at the end of the abstract section, you mentioned the ROC curves. However, you said nothing about these curves in the results. Please make conclusions only based on the results presented in the abstract.
Introduction
· “To date, there are few papers describing the features of MOC in MRI, mainly concerning tumor morphology” – please specify that you refer here to the quantitative features of MOCs.
Materials and Method:
· why only the Time To Peek and Perfusion Maximum Enhancement were the only parameters selected from the DCE? Why you did not consider other parameters such as kTRANS?
Results
· “The study included 66 women aged 33-78 years” – please provide the median + IQR / mean + standard deviation data for the age
· Table 4. Intraclass correlation of ADC parameter performed by the two radiologists – I suggest you should delete this table and instead integrate this information into the paragraph above.
· I am not sure if the tables’ format is the one required by the journal – please check this information and apply the necessary corrections
· The same observation as above regarding the figures and the figures’ legend
Discussion
· the first paragraph contains textbook knowledge. Also, you have already mentioned part of this information in the introduction. Please find a way to incorporate this information in the introduction and remove this paragraph
· you should start the discussion section by commenting on your results
· the paragraph starting with “Surgical treatment of MOC is likewise….” brings no relevant information related to your paper. Please remove any unnecessary information
· “MRI is the imaging modality with the highest tissue resolution. It is the method of choice for differentiation of primary ovarian tumor as well as diagnosing metastatic lesions.” please provide adequate citations for this affirmation
· the affirmation “We hope that this study is a contribution to further research on MOC.” has no role in the paragraph where you address the limitations. please remove this sentence
Conclusion
· “Those parameters show statistically significant differences between MOC, LGSC, and HGSC.” – which parameters?
· Please restrain the conclusions section to 4-5 lines.
Thank you very much for allowing me to express my opinions.
Sincerely,
P.
Author Response
Dear Reviewer
Thank you for your review and valuable comments. We have edited the text according to your remarks:
Simple summary:
- the sentence “This makes it easier to develop a treatment strategy.” is too. The results of this
study can, at best, increase confidence in the imaging diagnosis of MOC, but cannot decide the
therapeutic course. Please remove this statement.
The sentence “This makes it easier to develop a treatment strategy.” was removed from the text
Abstract:
- It should be mentioned in the abstract which area of the tumor was used to calculate the quantitative parameters. It may sound unnecessary, but some studies have measured these parameters both from the solid portion of the tumor and within the fluid portions. This should be clarified at the outset.
We have added a relevant sentence to the abstract: “ROI contained a small circle placed in the solid part of the primary tumor.”
- at the end of the abstract section, you mentioned the ROC curves. However, you said nothing about these curves in the results. Please make conclusions only based on the results presented in the abstract.
The ROC curves are presented in the Results section in subsection 3.4 and 3.5 thus we have mentioned them in the conclusions section
Introduction
- “To date, there are few papers describing the features of MOC in MRI, mainly concerning tumor morphology” – please specify that you refer here to the quantitative features of MOCs.
The sentence was changed accordingly: “To date, there are few papers describing morphological features of MOC in MRI, with no concern on the DCE and DWI parameters.”
Materials and Method:
- why only the Time To Peek and Perfusion Maximum Enhancement were the only parameters
selected from the DCE? Why you did not consider other parameters such as kTRANS?
- Thank you for this comment. We analyze different perfusion parameters such as K-trans, Kep, Ve. However, according to the literature and our experience, perfusion parameters may be variable, depending on the tumor grade. In EOC K-trans are variable for different FIGO stages. Higher K-trans values correlate with the aggressiveness of the tumor. In our study we have analyzed selected perfusion parameters, not recognized well in the context of MOC (such as Time To Peak and Perf. Max. En.), which may implicate the differential diagnosis between MOC and serous ovarian cancers. Application and analysis of other perfusion parameters is beyond the scope of this study. Our next prospective study in patients with ovarian cancer focuses on the MR imaging of the primary disease and K-trans as well as Ve will be evaluated, as those parameters correlate with the FIGO stage [Lindgren A, Anttila M, Arponen O, Rautiainen S, Könönen M, Vanninen R, Sallinen H. Prognosti value of preoperative dynamic contrast-enhancement magnetic resonance imaging in epithelial ovarian cancer. Eur J Radiol. 2019; 115:66-73]
Results
- “The study included 66 women aged 33-78 years” – please provide the median + IQR / mean +standard deviation data for the age
The values are as follows:
Mean ±SD 55.94 ±11.68
Median [Q1; Q3] 57.5 [48.5; 64]
IQR = Q3-Q1 = 15.5
Median and IQR values were added to the results section.
- Table 4. Intraclass correlation of ADC parameter performed by the two radiologists – I suggest you should delete this table and instead integrate this information into the paragraph above.
Thank you for the suggestion, table 4 was deleted from the text. We left the graph with inter-observer agreement.
Discussion
- The first paragraph contains textbook knowledge. Also, you have already mentioned part of this information in the introduction. Please find a way to incorporate this information in the introduction and remove this paragraph
We changed the text accordingly, and incorporated the part of the first paragraph from discussion to the introduction
- you should start the discussion section by commenting on your results
Thank you for the comment. We changed the text accordingly. The discussion section begins as follows:
“The presented results allowed us to determine the characteristics of MOC in DWI and DCE MRI. Our study showed that applying parameters such as ADC, TTP and Perf. Max. En. enables to differentiate MOC and serous EOC and dividing the latter into HGSCs and LGSCs.”
The paragraph starting with “Surgical treatment of MOC is likewise….” brings no relevant information related to your paper. Please remove any unnecessary information
Thank you for the comment. The paragraph about surgical treatment of MOC was removed from the text. We do agree that it brings no relevant information for the study.
- “MRI is the imaging modality with the highest tissue resolution. It is the method of choice for differentiation of primary ovarian tumor as well as diagnosing metastatic lesions.” please provide adequate citations for this affirmation
The adequate citation was added.
Portelance L, Corradini S, Erickson B, Lalondrelle S, Padgett K, van der Leij F, van Lier A and Jürgenliemk-Schulz I Online Magnetic Resonance-Guided Radiotherapy (oMRgRT) for Gynecological Cancers. Oncol 2021; 11:628131. Alves AS, Felix
A, Cunha TM. Clues to the diagnosis of borderline ovarian tumours: An imaging guide. Eur J Radiol. 2021; 143: 109904
Ohya A, Fujinaga Y. Magnetic resonance imaging findings of cystic ovarian tumors: major differential diagnoses in five types frequently encountered in daily clinical practice. Jpn J Radiol. 2022; 40: 1213-1234
Taylor EC, MD Lina Irshaid, MD Mahan Mathur, MD Multimodality Imaging Approach to Ovarian Neoplasms with Pathologic Correlation RadioGraphics 2021; 41:289–315
- the affirmation “We hope that this study is a contribution to further research on MOC.” has no role in the paragraph where you address the limitations. please remove this sentence
Thank you for the comment. We removed the sentence from the text.
Conclusion
- “Those parameters show statistically significant differences between MOC, LGSC, and HGSC.” – which parameters?
- Please restrain the conclusions section to 4-5 lines.
The conclusions section was shortened and rearranged as follows:
DWI and DCE seem to be very good diagnostic tools for differentiating between serous carcinomas (LGSC, HGSC) and MOC. Significant, differences in median ADC values between MOC and LGSC and between MOC and HGSC demonstrate the usefulness of DWI in differentiating between less and more aggressive types of EOC, not only in the group of the most common serous carcinomas.
The analysis of the ROC curves showed that ADC had excellent diagnostic accuracy in differentiating between MOC and HGSC. While in case of differentiation between LGSC and MOC, TTP showed the greatest differentiating value.
Thank you for the review.
Reviewer 4 Report
In general, this manuscript is quite lengthy and should be substantially shortened to be clearer and more easily understandable. For the same reasons it should also undergo extensive English language editing, preferably by a native English speaking medical writer or a professional editing service.
In addition:
1) The title seems a list of the topics addressed in the study, illustrated by 3 brief independent statements separated by periods. It should be modified into a single or, at most, two sentences linked together and focused on the main study goal.
2) The Discussion should be much more focused on commenting upon and hypothesizing the pathophysiological and clinical bases for the study findings, instead of just repeating them or expanding generic information on the diagnostic and therapeutic workup of MOC.
Author Response
Thank you for your review and valuable comments. We have edited the text according to your remarks:
1) The title seems a list of the topics addressed in the study, illustrated by 3 brief independent statements separated by periods. It should be modified into a single or, at most, two sentences linked together and focused on the main study goal.
Thank you for the comment, the title was changed as follows:
“Characterization of primary mucinous ovarian cancer by diffusion-weighted and dynamic contrast enhancement MRI and comparison with serous ovarian cancer. ”
2) The Discussion should be much more focused on commenting upon and hypothesizing the pathophysiological and clinical bases for the study findings, instead of just repeating them or expanding generic information on the diagnostic and therapeutic workup of MOC
The discussion section was shortened and the paragraph about surgical treatment of MOC was deleted. The introduction was rearranged and shortened accordingly.
Additionally, table 4 with interobserver agreement data was removed from the text as well.
In line with your suggestions about language corrections, the text was edited by native English-speaking doctor who helped us with our previous study published in Cancers Journal last year.
Thank you for your review.
Round 2
Reviewer 3 Report
The autohors adreesed all of my concerns.
Reviewer 4 Report
Thank you. No further comments.